# Predictors and number of antenatal care visits among reproductive age women in Sub-Saharan Africa further analysis of recent demographic and health survey from 2017–2023: Zero-inflated negative binomial regression

**Abel Endawkie**[1]*, **Natnael Kebede**[2], **Desale Bihonegn Asmamaw**[3], **Yawkal Tsega**[4]

1 Department of Epidemiology and Biostatistics, School of Public Health, College of Medicine and Health Sciences, Wollo University, Dessie, Ethiopia, 2 Department of Health Promotion, School of Public Health College of Medicine and Health Sciences, Wollo University, Dessie, Ethiopia, 3 Department of Reproductive Health, Institute of Public Health, College of Medicine and Health Sciences, University of Gondar, Gondar, Ethiopia, 4 Department of Health System and Management, School of Public Health, College of Medicine and Health Sciences, Wollo University, Dessie, Ethiopia

* abelendawkie@gmail.com

**Data Availability Statement:** Data is available in a public, open access repository. Data for this study were sourced from Demographic and Health

## Abstract

### Background

Antenatal care (ANC) is an important component of maternal and child health care. To reduce perinatal mortality and improve women's experience of care, the World Health Organization (WHO) recommends pregnant women should attend at least eight ANC visits. However, in Sub-Saharan Africa (SSA), the mean number of ANC visits among reproductive-age women using Demographic and Health Surveys (DHS) data following the new WHO recommendation is so far limited. Therefore, the study aimed to determine the mean number of ANC visits and its predictors among reproductive-age women in SSA.

### Method

Community-based cross-sectional study was conducted among 188,880 weighted reproductive-age women in SSA using a recent round of DHS data from 2017–2023. Zero-inflated negative binomial regression (ZINB) was conducted and statistical significance was declared at p-value <0.05 and adjusted incidence rate ratio(AIRR) for count model part and adjusted odds ratio for logit model inflated part of ZINBR with 95% confidence interval (CI) were reported.

### Result

The mean number of ANC visits among reproductive-age women in SSA was 4.08 with 95% CI [4.07, 4.09]. Among reproductive-age women who gave birth in the last five years before the survey, 7.3% had eight or more ANC visits during pregnancy. Age of women, maternal

surveys (DHS) and available here: https://dhsprogram.com/.

**Funding:** The author(s) received no specific funding for this work.

**Competing interests:** The authors declare that they have no competing interests.

**Abbreviations:** AIC, Akakian information Criteria; AIRR, Adjusted incidence rate ratio; ANC, Antenatal care; BIC, Bayesian information Criteria; CSA, central statistical agency; DHS, Demographic and Health Survey; EA, Enumeration Area; LR, log-likelihood Ratio; NBR, negative binomial regression; SSA, Sub-Saharan Africa; WHO, World Health Organization; ZINBR, Zero-inflated negative binomial regression.

and husband educational status, types of pregnancy, birth order, household size, number of under-five children, and wealth index were associated with the numbers of ANC visits among reproductive-age women in Sub-Saharan.

## Conclusion

The mean number of ANC visits among reproductive-age women in SSA is too lower than the new WHO recommendation of ANC visits for a positive pregnancy experience. This study also highlights that the proportion of at least eight ANC visits is low and there are still disparities in the mean of ANC visits across different regions of SSA. The increasing maternal age, higher maternal and husband educational status, wanted pregnancy, the number of household members, the number of under-five children, and higher wealth index increase the number of ANC visits. Unwanted pregnancy, no more fertility desire, and rural residences were contributed for zero ANC visits in SSA. Therefore, efforts should be geared towards improving maternal and husband's educational status. We strongly recommend that the governments of SSA countries should empower women economically and educationally to achieve the goals of ANC as recommended by the WHO.

## Introduction

Antenatal care (ANC) is an important component of maternal and child health care, aimed at improving the health of pregnant women and their unborn babies [1]. It is a sequence of clinical tests and interventions that are aimed at ensuring the well-being of the pregnant mother and her fetus [2,3]. The WHO recommends pregnant women should attend a minimum of eight ANC visits to reduce perinatal mortality and improve women's experience of care [4]. Antenatal care constitutes vital healthcare service for the duration of pregnancy, delivered both at a healthcare facility and inside the home, gambling a pivotal position in maternal and child fitness [4,5]. Globally, approximately 295,000 women died from preventable causes related to pregnancy and childbirth, an average of 810 deaths per day [6]. Most of these deaths (94%) occurred in areas where resources were limited and many could have been prevented [6]. Sub-Saharan Africa accounted for more than two-thirds (196,000) of all maternal deaths globally [6]. The global maternal mortality rate (MMR), which refers to maternal deaths per 100,000 live births, is about 216, with a massive 95% in developing countries [7]. Despite the recent launch of Sustainable Development Goals (SDGs) target 3.1 aimed to reduce maternal mortality (MMR) to 70 per 100,000 live births globally by 2030 [8], MMR remains a significant public health challenge in SSA [9]. Many pregnant women in Africa do not receive ANC services [5,10,11] and underutilization of ANC visits may result in adverse maternal and neonatal outcomes, including maternal morbidity and mortality, preterm delivery, low birth weight, and neonatal mortality [6,7]. According to a different study finding, the factors of ANC utilization were maternal age [12–14] educational status of women [15–17], marital status of the woman [16,18], husband's occupation [18,19] and level of education [18,20], media exposure [16,21], household wealth index [18,22], community women's literacy [23], interval between births [15,24], first birth [25] transportation issues [24,26], and distance to the health facility [24,27]. This study showed inconsistent findings in associations of predictors with ANC visits. For example, the study conducted by Ahinkorah BO, et.al in Guinea [14] showed that women who are aged 15–24 years were more likely to receive ANC whereas a study conducted by

Azanaw MM, et.al in Ethiopia [13] and Tessema ZA, et.al at SSA [12] revealed that women who are aged 15–24 years were less likely to receive ANC. Likewise, there is a discrepancy of findings for women with their first child were less likely to receive ANC services, and women with their first child were more likely to receive ANC services [13,25,28]. Moreover, there is extensive evidence of the prevalence and associated factors for ANC visits among reproductive-age women [16,20,22,25,29–34] and limited evidence related to mean number of ANC visits and its predictors among reproductive age women using count model [35–37] in different countries of SSA. However, there is paucity of evidence related to mean number of ANC visits and its predictors among reproductive age women using count model in SSA using recent DHS data following the new WHO recommendation [4]. Moreover, investigating using the count model help to improve information loss and provides the average number of ANC visits and its predictors among reproductive age women. This study highlights the proportion of reproductive age women who were utilizing at least eight ANC visits and also establish evidence for policy evaluators on the ANC utilization in SSA countries based on the new WHO recommendation of ANC visit for a positive pregnancy experience [4]. Therefore, the study aims to determine the mean number of ANC visits, proportion of at least eight ANC visits and its predictors among reproductive age women in Sub-Saharan Africa using recent DHS data from 2017–2023.

## Method

### Study design

The research used a cross-sectional study design used in the recent DHS in Sub-Saharan Africa.

### Study settings and population

The study was done in Sub-Saharan Africa, which is located in Continent Africa, with a diverse population [38]. SSA or Non-Mediterranean Africa is the area and regions of the continent of Africa that lie south of the Sahara [39]. These include Central Africa (Burundi, Cameron, Gabon, Guiana, and Cotdivore) East Africa (Ethiopia, Kenya, Tanzania, and Rwanda) Southern Africa(Madagascar, Mali, Zambia, and Muartie), and West Africa((Burkina Faso, Gambia, Nigeria, Liberia, Senegal, and Seriee Lion). It covers a large area with over 40 countries, about 1 billion people, and a rich and diverse culture. The study was conducted using the most recent nationally representative cross-sectional DHS data available from 2017–2023 following the 2016 new WHO recommendation of ANC visit for a positive pregnancy experience [4].

### Source and study population

The source population was all reproductive-age women who were pregnant and gave birth five years before the survey in Sub-Saharan Africa, whereas those in the selected Enumeration Areas (EAs) were the study population.

**Data source.**   We extracted the recent rounds of DHS data of 19 SSA countries. These countries were subdivided by region: Central Africa (Burundi, Cameron, Gabon, Guiana, and Cotdivore), East Africa (Ethiopia, Kenya, Tanzania, and Rwanda), Southern Africa (Madagascar, Mali, Zambia, and Muartie), and West Africa ((Burkina Faso, Gambia, Nigeria, Liberia, Senegal, and Sere Lion) and the details of country and years of DHS for this study is provided as (S1 File). The data set was birth record (BR) which contains the full birth history of all women interviewed. The data set also included information on pregnancy and postnatal care, as well as immunization, health, and nutrition data for children born in the last 5 years [40,41].

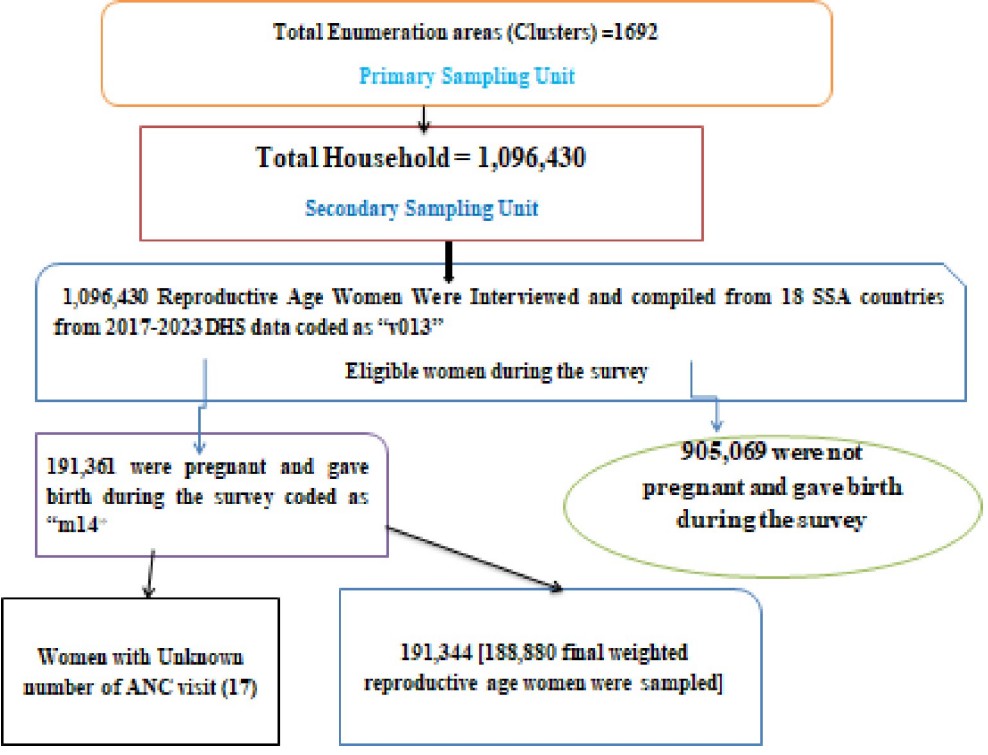

**Fig 1. Sampling and exclusion procedures to identify the final sample size of the study.**

The DHS was a household survey that was conducted every five years in low and middle-income countries. It is a crucial data source for maternal health care utilization issues in these countries, as it contains data on various reproductive health issues such as marriage, antenatal care visits, postnatal visits, fertility preferences, and contraception [40]. The data collected from the DHS survey was organized in a hierarchical structure, with households within a cluster forming the top level. The next level consists of household members, followed by interviewed women and men as a subset of household members. The bottom levels of the hierarchy include pregnancies and children of each interviewed woman [40,41].

**Sample size and sampling method.** The sample size was determined from the birth record file 'BR' from the DHS dataset of Sub-Saharan African countries that had at least one survey conducted between 2017 and 2023. DHS uses a two-stage stratified cluster sampling technique. In the first stage, a sample of enumeration areas (EAs) was selected independently from each stratum with proportional allocation stratified by residence (urban and rural). In the second stage, households were taken from the selected EAs using a systematic sampling technique [41]. The final weighted sample size was 188,880 reproductive-age women were included (Fig 1).

## Inclusion and exclusion criteria

Those who were pregnant and gave birth in the last 5 years before the survey and whose status of ANC visit was known were included in the study. The study also included women who did not get ANC during their pregnancy. Nevertheless, the study did not include women documented as having pregnancy termination.

**Study variables.** The dependent variable of this study was the number of antenatal care visits among pregnant women from 0–9/10 months of pregnancy until the data collection period counted as (0, 1, 2, 3, 4, 5, 6, 7, 8, 9, 10. . . . . . 20). The independent variables included socio-demographic and economic-related factors like (individual maternal age, educational status, marital status, religion, partner educational status, household level factors like; sex of the household head, age of the household head, household wealth index, community level factors like; residence, community level wealth index, community level educational status, region of SSA) and obstetric factors like birth interval, types of pregnancy, previous cesarean section were independent variable.

## Measurement and operational definitions

**Count regression.** The count regression model used for the non-negative integer which was counted data as 0, 1, 2, 3. . . [42].

**Antenatal care visit.** The number of antenatal care visits a woman received during the last pregnancy before the survey in this study was counted as a non-negative integer using a proper count regression model as 0,1,2,3,4. . . . . . . . ..20.

## Data processing and analysis

Data used were extracted, cleaned, coded, and analyzed using STATA version 17 Statistical software. Sample weights were done before further analysis, and descriptive statistics were described using frequencies, percentages, mean, and standard deviation, and presented using tables, figures, and narratives.

**Poisson regression model.** The Poisson regression model is a statistical technique used to count data as a function of a set of independent variables. It is often used to model the relationship between a count response variable and one or more predictor variables. The model assumes that the response variable follows a Poisson distribution and that the logarithm of its expected value can be modeled as a linear combination of the predictor variables. The standard Poisson regression model assumes that the observations are independent over time and that the mean and variance of the dependent variable are equal [42]. Since the variance exceeds the mean (variance = 6.55 and mean = 4.08) in this study there is over-dispersion and the premise of the Poisson regression has failed. Therefore, the negative binomial regression model with an unobserved particular effect (random term or error term) for the parameter was chosen to account for the over-dispersion. The over-dispersion parameter in the negative binomial (NB) specification was tested using a Likelihood Ratio (LR) test for the parameter α of p-value < 0.001 in contrast to the Poisson model specification in addition to the comparison of mean and variance [43].

**Multilevel mixed effect negative binomial regression model.** Since DHS data is hierarchical [44] it assumes the reproductive-age women and their households are nested within enumeration areas (communities). This suggests that reproductive-age women in households with similar characteristics can have different health outcomes when residing in different communities with different characteristics. Therefore, to check whether there are distinct measures of variation and any potential clustering effect for conducting multilevel analysis: the intra-class correlation coefficient (ICC) was calculated using the following formula [45].

ICC $= \frac{\delta 2}{\delta 2 + \pi 2/3}$ where δ2 indicates the estimated constant variance of clusters in the null model (obtained from variance components of the null model or empty model) and π2/3 is the variance of the model with scale parameter (residual variance) which is approximately 3.29. The calculated ICC value using the above formula was 3.2% which is less than 10% even 5% which favors the individual-level analysis using negative binomial regression analysis.

**Negative binomial regression.** Due to the over-dispersion of the count data (variance greater than the mean), the negative binomial regression was considered [46]. To detect zero inflation due to excess zero [47], the Vuong test was conducted. In this data, the Vuong test of zero-inflated negative binomial regression and standard negative binomial regression (Vuong test of zip vs. standard Poisson: z = 65.46 and p-value = 0.0000) which indicates that zero-inflated negative binomial regression is better than the standard negative binomial regression model for this data.

**Zero-inflated negative binomial regression.** Zero-inflated negative binomial regression is a statistical technique employed when working with count data that exhibits excessive zeros and over-dispersion beyond what a standard negative binomial regression predicts. The excessive zeros do not exactly mean there are excessive zeros at all in frequency distribution rather the model is affected by the presence of excessive zeros or structural zeros [48]. This method consists of two components: the first is a count model that predicts the predictors of the actual count occurrence (frequency of non-zero count) and a logit model predicts the predictors for the occurrence of excess zeros or it estimates the probability of observing a zero count (excess zeros) in the data [49].

After the bivariable analysis was performed, variables having a p-value less than 0.2 were entered into a multivariable zero-inflated negative binomial regression model to determine the predictors of the mean number of ANC visits during pregnancy in SSA using the recent round of DHS data from 2017–2023 by accounting both excess zero and over-dispersion.

However, the Vuong test is not sufficient, and traditional criteria to decide the model [50], the multivariable zero-inflated negative binomial regression, and the standard multivariable negative binomial regression model were compared using Akakian Information Criteria (AIC)/ Bayesian Information Criteria (BIC). The multi-colinearity was checked using the variance inflation factor (VIF), and the fitted model's mean VIF was 1.5. Statistical significance was declared at p-value <0.05 and adjusted incidence rate ratio(AIRR) for the count model part and adjusted odds ratio for the logit model (inflated) part of ZINBR with 95% confidence interval (CI) were reported.

*Ethical approval.* No ethical approval was needed because we had used the demographic and health survey which de-identifies all data before making it public, and the used DHS data sets are openly accessible. An authorization letter was requested to download the DHS data set and this was obtained from the Central Statistical Agency (CSA) after being requested at https://dhsprogram.com/. The dataset and all methods of this study were conducted according to the guidelines laid down in the Declaration of Helsinki and based on DHS research guidelines.

## Result

### Socio-demographic characteristics with pairwise mean number of ANC visits

The study included a weighted sample of 188,880 women of reproductive age. The mean age and standard deviation of these women were 29 ±7 respectively. Among the total number of women, 69,840 (36.4%) who were not educated had a mean number of 3.39 antenatal care (ANC) visits. On the other hand, 9,044 (4.9%) women of reproductive age with higher education had a mean number of 5.94 ANC visits. The mean number of ANC visits was lower among women who lived in the poorest household wealth index (3.29) than those in the richest household wealth index (5.08). Urban dwellers had a higher mean number of ANC visits (4.7) than rural dwellers (3.68). Women of reproductive age in Central Africa, West Africa, East

Africa, and South Africa had a mean number of ANC visits of 3.79, 4.11, 4.00, and 3.98 respectively (Table 1).

## Mean numbers of ANC visits among reproductive-age women in SSA

The average number of antenatal care (ANC) visits among women of reproductive age in Sub-Saharan Africa was 4.08 ±2.56 standard deviation with 95% CI [4.07, 4.09]. Out of 188,880 reproductive-age women who were pregnant and gave birth five years before the survey 7.3% had eight or more ANC visits during pregnancy and 9.2% had no ANC visit (Table 2). The mean number of ANC visits was lowest in Central Africa (3.79) and highest in West Africa (4.11).

## Predictors of the number of ANC visits among reproductive-age women in SSA using zero-inflated negative binomial regression analysis

Different count models were fitted for model selection, and because multivariable zero-inflated negative binomial regression accounts for the over-dispersion and excess zero, the multivariable zero-inflated negative binomial regression model has the lowest AIC and BIC value than the standard negative binomial regression model. The AIC value for the Zero-inflated negative binomial regression was 92,777, while that of the standard negative binomial regression model was 94,818 (Table 3). In this study, maternal age, women and husband educational status, types of pregnancy, birth order, household size, number of under-five children, and household wealth index were statistically significant at p-value <0.05 with frequency of ANC visits.

The frequency of ANC visits were 1.04 [AIRR 1.04, 95%CI (1.01, 1.17)], 1.06 [AIRR 1.06, 95%CI (1.03, 1.09)], 1.09 [AIRR 1.09, 95% CI (1.05, 1.13)], 1.11 [AIRR 1.11, 95% CI (1.07, 1.15)], 1.15 [AIRR 1.15, 95% CI (1.1, 1.2), and 1.15 times (AIRR 1.15, 95% CI (1.07, 1.24) for women aged 20–24, 24–29, 30–34, 35–39, 40–44, and 45–49 more likely as compared to women of 15–19 years of age respectively. The number of ANC visits was higher for women who had primary [AIRR, 1.04, 95% CI (1.02, 1.055)], secondary [AIRR, 1.04, 95% CI (1.02, 1.06)], and higher education [AIRR 1.08, 95% CI (1.04, 1.19)] as compared to women who have no education respectively. Women with partners who have primary, secondary, and higher education had a higher frequency of ANC visits [AIRR, 1.05, 95% CI (1.04, 1.07)], [AIRR, 1.06, 95% CI (1.04, 1.08)] and [AIRR, 1.08, 95% CI (1.05, 1.12)] as compared with women with a partner who have no education respectively. The frequency of ANC visits among women who have unwanted pregnancies decreased by 9% as compared to wanted pregnancies. The frequencies of ANC visits among women decreased by 2.9% when the birth orders were increased [AIRR 0.971, 95% CI (0.97, 0.98)]. The frequencies of ANC visits among women were increased by 1% when the household size was increased [AIRR 1.01, 95% CI (1.00, 1.011)]. The frequency of ANC visits among women decreased by 2.7% when the number of under-five children was increased [AIRR 0.973, 95% CI (0.97, 0.98)]. The frequency of ANC visits was higher among women from poorer [AIRR, 1.07, 95% CI (1.05, 1.1)], middle [AIRR, 1.09, 95% CI (1.07, 1.11)], richer [AIRR, 1.14, 95% CI (1.12, 1.17)], and richest [AIRR, 1.27, 95% CI (1.23, 1.30)] as compared with women from poorest household wealth index respectively (Table 4).

The odds of zero ANC visits were more likely among pregnant women whose ages in between 30–34 and 45–49 (AOR, 1.55 (1.01, 2.36) and (AOR, 2.13(1.09, 4.17), who had primary education (AOR, 1.43 (1.18, 1.74), unwanted pregnancy (AOR, 1.6(1.16, 2.2), no more fertility preference (AOR, 1.32(1.1, 1.57) and who were from rural residence (AOR, 1.42 (1.1,1.84) as compared to their counterpart. The odds of zero ANC visits were less likely among mothers who had secondary education (AOR, 0.71(0.53, 0.96) and occupation (AOR,

**Table 1. Socio-demographic characters and pairwise comparisons of means of ANC visits among reproductive-age women in SSA using recent rounds of DHS from 2017–2023.**

| Variable | Frequency | Percentage | Mean# of ANC visit with 95%CI |
|---|---|---|---|
| Maternal age | | | |
| 15–19 | 13,775 | 7.4 | 3.74(3.7,3.78) |
| 20–24 | 41,107 | 22.0 | 3.99(3.96,4.01) |
| 25–29 | 47,359 | 25.4 | 4.07(4.04,4.09) |
| 30–34 | 38,107 | 20.4 | 4.12(4.09,4.15) |
| 35–39 | 28,678 | 15.4 | 4.10(4.08,4.13) |
| 40–44 | 13,317 | 7.1 | 3.93(3.883.97) |
| 45–49 | 4,263 | 2.3 | 3.62(3.55,3.67) |
| Maternal education | | | |
| No education | 69,840 | 37.4 | 3.39(3.37,3.4) |
| Primary education | 57,871 | 31.0 | 3.89(3.87,3.91) |
| Secondary education | 49,851 | 26.7 | 4.83(4.8,4.85) |
| Higher education | 9,044 | 4.9 | 5.94(5.88,5.99) |
| Maternal occupation status | | | |
| No have work | 67,612 | 37.0 | 3.80(3.78,3.82) |
| yes have work | 115,077 | 63.0 | 4.20(4.18,4.21) |
| Partner(husband) education | | | |
| No education | 63,167 | 40.9 | 3.39(3.38,3.41) |
| Primary education | 40,584 | 26.3 | 3.85(3.83,3.88) |
| Secondary education | 38,121 | 24.7 | 4.83(4.81,4.86) |
| Higher education | 12,740 | 8.2 | 5.58(5.53,5.62) |
| Husband occupation status | | | |
| No have work | 15,954 | 10.2 | 3.60(3.56,3.64) |
| Yes have work | 140,460 | 89.8 | 4.07(4.06,4.09) |
| Pregnancy type | | | |
| Wanted | 30,814 | 95.3 | 4.01(4,4.02) |
| Unwanted | 1,538 | 4.8 | 3.5(3.4,3.6) |
| fertility Preference | | | |
| Prefer | 120,982 | 67.9 | 4.06(4.05,4.08) |
| Not prefer | 57,128 | 32.1 | 4.00(3.98,4.02) |
| Media exposure | | | |
| Not Exposed | 94,346 | 51.6 | 3.66(3.65,3.68) |
| Exposed | 88,340 | 48.4 | 4.50(4.48,4.51) |
| Birth order | | | |
| 1 | 41,424 | 22.7 | 4.28(4.26,4.31) |
| 2 | 36,668 | 20.1 | 4.19(4.16,4.22) |
| 3 | 30,316 | 16.6 | 4.14(4.11,4.17) |
| 4 | 23,273 | 12.7 | 4.04(4.01,4.07) |
| 5 | 17,230 | 9.4 | 3.96(3.93,4) |
| 6 | 12,615 | 6.9 | 3.75(3.7,3.79) |
| 7 | 9,005 | 4.9 | 3.67(3.62,3.73) |
| 8 | 5,542 | 3.0 | 3.53(3.47,3.6) |
| 9 | 3,071 | 1.7 | 3.44(3.35,3.52) |
| 10+ | 3,545 | 1.9 | 3.06(2.98,3.14) |
| Sex of household head | | | |
| Male | 148,557 | 79.6 | 3.97(3.96,3.99) |

(*Continued*)

**Table 1.** (Continued)

| Variable | Frequency | Percentage | Mean# of ANC visit with 95%CI |
|---|---|---|---|
| Female | 38,048 | 20.4 | 4.20(4.18,4.23) |
| Age of household head | | | |
| 15–20 | 1,422 | 0.9 | 3.68(3.57,3.79) |
| 21–30 | 38,742 | 25.1 | 4.01(3.99,4.04) |
| 31–40 | 63,680 | 41.2 | 4.11(4.09,4.13) |
| 41–50 | 35,790 | 23.1 | 4.04(4.02,4.07) |
| 51–64 | 12,667 | 8.2 | 3.98(3.95,4.01) |
| >64 | 2,378 | 1.5 | 4.01(3.97,4.05) |
| Wealth index of household | | | |
| Poorest | 40,582 | 21.8 | 3.29(3.27,3.31) |
| Poorer | 38,814 | 20.8 | 3.71(3.69,3.74) |
| Middle | 37,252 | 20.0 | 4.05(4.03,4.08) |
| Richer | 36,709 | 19.7 | 4.44(4.42,4.47) |
| Richest | 33,249 | 17.8 | 5.11(5.08,5.14) |
| Residence | | | |
| Urban | 67,040 | 35.9 | 4.70(4.68,4.72) |
| Rural | 119,566 | 64.1 | 3.68(3.67,3.7) |
| Region of SSA | | | |
| Central Africa | 18,275 | 9.8 | 3.79(3.75,3.82) |
| West Africa | 78,407 | 42.0 | 4.11(4.09,4.12) |
| East Africa | 39,761 | 21.3 | 4.00(3.98,4.03) |
| South Africa | 50,163 | 26.9 | 3.98(3.96,4) |

0.38(0.32, 0.45) and whose husbands had occupation (0.59(0.49, 0.710,) media exposure (AOR, 0.62(0.52, 0.75) and from poorer (AOR, 0.80(0.65, 0.99), middle (AOR, 0.57(0.44, 0.74) and richer (AOR, 0.75 (0.57, 0.97) as compared to poorest household(Table 4).

**Table 2. Number of ANC visits among reproductive-age women experiencing SSA using 2017–2023 DHS.**

| #of ANC visits during pregnancy | Number of women who experienced ANC visit | Percent |
|---|---|---|
| 0 | 17,172 | 9.2 |
| 1 | 7,460 | 4 |
| 2 | 14,755 | 7.91 |
| 3 | 35,241 | 18.89 |
| 4 | 44,744 | 23.98 |
| 5 | 26,478 | 14.19 |
| 6 | 18,824 | 10.09 |
| 7 | 8,351 | 4.48 |
| 8 | 5,983 | 3.21 |
| 9 | 2,553 | 1.37 |
| 10+ | 5,043 | 2.7 |
| | The mean number of ANC visits | |
| Mean | 4.08, 95%CI (4.07,4.09) | |
| Standard deviation | 2.56 | |
| Minimum | 0 | |
| Maximum | 20 | |

**Table 3. Model selection.**

| Criteria | Model (NBR) | (ZINBR) |
|---|---|---|
| LR | -47377 | -46353 |
| AIC | 94818 | 92777 |
| BIC | 95078 | 93061 |

**Key:** NBR = negative binomial regression, ZINBR = Zero-inflated negative binomial regression, LR = log likelihood Ratio, AIC = Akakian information Criteria, BIC = Bayesian information Criteria.

## Discussion

This research was conducted as a statistical analysis using zero-inflated negative binomial regression to determine the average number of antenatal care (ANC) visits and its predictors among 188,880 reproductive-age women in Sub-Saharan Africa using recent rounds of DHS data from 2017–2023. Out of 188,880 reproductive-age women who were pregnant and gave birth in the last five years before the survey 7.3% had eight or more ANC visits during pregnancy which is slightly higher than previous study findings [29]. The mean number of ANC visits among reproductive-age women in Sub-Saharan Africa was 4.08, 95%CI [4.07, 4.09] which was slightly higher than the previous study findings in Ethiopia [35], and in SSA(3.83) [36] and lower than the recent study findings in the low-and middle-income countries [24]. This indicates a minor improvement in the frequency of ANC visits. However, it is below the WHO's updated recommendation for ANC visits for a positive pregnancy experience in SSA [4]. This indicates that most reproductive-age women in Sub-Saharan African countries were not getting minimum ANC visits during pregnancy based on the new WHO's guideline on ANC visits for a positive pregnancy experience [4]. The WHO recommends that a woman should get at least eight ANC visits during pregnancy for a positive pregnancy experience. This study also found that there are still disparities in the mean number of ANC visits across different regions of SSA. These disparities could be due to inequalities in the accessibility of maternal health services, poor or absent transportation, inequality in the number of health care providers, and disparities in access to education. Additionally, the differences in the implementation of maternal and child health programs, women's education, and the role of women in household wealth status across different regions of SSA could also contribute to these disparities.

In this study, maternal age, women and husband educational status, types of pregnancy, fertility preference, birth order, household size, number of under-five children, and household wealth index, were predictors of the numbers of ANC and zero ANC visits which are similar to findings from previous study [16–19,30,51,52]. The frequency of ANC visits were 1.04, 1.06, 1.09, 1.11, 1.15, and 1.15 times more likely for women aged 20–24, 24–29, 30–34, 35–39, 40–44, and 45–49 as compared to women of 15–19 age respectively. The higher age of pregnant women was positively associated with the number of ANC visits. This indicates that older pregnant women are more likely to schedule to get more numbers of ANC visits. This is similar to findings from other studies [35,36]. This may be due to the fact that as women age increase, they may experience birth-related complications and poor health conditions, which could lead them to demand more visits. Additionally, older women may have more experience with pregnancy care compared to younger women (15–19 years old). Correspondingly the higher and lower educational status of pregnant women and their husbands had a positive association with the frequency of ANC and zero ANC visits respectively. This was supported by other previous study findings [11,17,23,30,36]. This may be due to the fact that educated husbands may have a better ability to convince their spouses to have more prenatal care during

**Table 4. Zero-inflated negative binomial regression analysis for predictors of the number of ANC visits in SSA (n = 188,880).**

| Variable | Mean# of ANC visits | Model 1(NBR) | Model 2 (Zero-inflated negative binomial regression (ZINBR) | |
|---|---|---|---|---|
| | | AIRR with 95% CI | AIRR With 95% CI | The inflated part of ZINBR (AOR with 95% CI) |
| Maternal age | | | | |
| 15–19 | 3.74 | Reference | | |
| 20–24 | 3.99 | 1.03(1.00,1.06) | 1.04(1.01,1.07)** | 1.32(0.92, 1.91) |
| 25–29 | 4.07 | 1.05(1.02,1.09) | 1.06(1.03,1.09)*** | 1.33(0.90, 1.96) |
| 30–34 | 4.12 | 1.08 (1.04,1.11) | 1.09(1.05,1.13)*** | 1.55 (1.01, 2.36)* |
| 35–39 | 4.10 | 1.10 (1.06,1.14) | 1.11(1.07,1.15)*** | 1.4 (0.88, 2.24) |
| 40–44 | 3.93 | 1.14(1.09,1.19) | 1.15(1.10,1.20)*** | 1.29(0.75, 2.25) |
| 45–49 | 3.62 | 1.11(1.03,1.19) | 1.17(1.07,1.24)*** | 2.13(1.09, 4.17)* |
| Maternal education | | | | |
| No education | 3.39 | Reference | | |
| Primary education | 3.89 | 1.015(0.99,1.03) | 1.03(1.02,1.06)*** | 1.43 (1.18, 1.74)* |
| Secondary education | 4.83 | 1.047(1.02,1.07) | 1.04(1.02,1.06)*** | 0.71(0.53, 0.96) |
| Higher education | 5.94 | 1.092(1.05,1.14) | 1.07(1.04,1.12)*** | 0.00001(0.0,35.2) |
| Maternal occupation status | | | | |
| Have no work | 3.80 | Reference | Reference | |
| Have work | 4.20 | 1.04 (1.03,1.06) | 1.01(0.99,1.030) | 0.38(0.32, 0.45)* |
| Partner(husband) education | | | | |
| No education | 3.39 | Reference | Reference | |
| Primary education | 3.85 | 1.02(1.00, 1.04) | 1.05(1.04,1.07)*** | 1.89(1.54, 2.33)* |
| Secondary education | 4.83 | 1.04(1.02,1.06) | 1.06(1.04,1.08)*** | 1.34(1.01, 1.76)* |
| Higher education | 5.58 | 1.08(1.05,1.12) | 1.08(1.05,1.12)*** | 0.43(0.176, 1.04) |
| Husband occupation status | | | | |
| No have work | 3.60 | Reference | Reference | |
| Yes have work | 4.07 | 1.01(0.99,1.03) | 0.99(0.96,1.01) | 0.59(0.49, 0.71)* |
| Pregnancy type | | | | |
| Wanted | 4.0 | Reference | Reference | |
| Unwanted | 3.5 | 0.89(0.86,0.93) | 0.91(0.87,0.95)*** | 1.6(1.16, 2.20)* |
| Fertility Preference | | | | |
| Prefer | 4.06 | Reference | Reference | |
| Not prefer | 4.00 | 1.00 (0.98,1.01) | 1.01(0.99,1.02) | 1.32(1.10, 1.57)* |
| Media exposure | | | | |
| Not Exposed | 3.66 | Reference | Reference | |
| Exposed | 4.50 | 1.02(1.01,1.040 | 1.01(0.99,1.02) | 0.62(0.52, 0.75)* |
| Birth order | | 0.97(0.97,0.98) | 0.97(0.97,0.98)** | 1.012(0.96, 1.07) |
| Household size | | 1.01(1.00,1.01) | 1.01(1.00,1.01)*** | 0.99(0.95, 1.03) |
| Number of under-five children | | 0.975(0.97,0.98) | 0.97(0.97,0.98)*** | 0.95(0.86, 1.02) |
| Sex of household head | | | | |
| Male | 3.97 | Reference | Reference | |
| Female | 4.20 | 1.00(0.98,1.02) | 1.00(0.98,1.02) | 1.08(0.87, 1.35) |
| Age of household head | | | | |
| 15–20 | 3.68 | Reference | Reference | |
| 21–30 | 4.01 | 1.00(0.92,1.08) | 1.00(0.92,1.08) | 0.92(0.38, 2.11) |
| 31–40 | 4.11 | 1.00(0.93,1.09) | 1.00(0.92,1.09) | 0.77(0.33, 1.83) |
| 41–50 | 4.04 | 1.01(0.93,1.10) | 1.01(0.93,1.09) | 0.73(0.31, 1.75) |
| 51–64 | 3.98 | 0.95(0.88,1.03) | 0.96(0.88,1.04) | 1.12(0.47, 2.67) |
| >64 | 4.01 | 0.99(0.91,1.07) | 0.99(0.91,1.08) | 1.09(0.45, 2.66) |

*(Continued)*

**Table 4.** (Continued)

| Variable | Mean# of ANC visits | Model 1(NBR) | Model 2 (Zero-inflated negative binomial regression (ZINBR) | |
|---|---|---|---|---|
| | | AIRR with 95% CI | AIRR With 95% CI | The inflated part of ZINBR (AOR with 95% CI) |
| Wealth index of household | | | | |
| Poorest | 3.29 | Reference | Reference | |
| Poorer | 3.71 | 1.08(1.06,1.10) | 1.07(1.05,1.10)*** | 0.80(0.65, 0.99)* |
| Middle | 4.05 | 1.11(1.08,1.13) | 1.09(1.07,1.11)*** | 0.57(0.44, 0.74)* |
| Richer | 4.44 | 1.15(1.13,1.18) | 1.14(1.12,1.17)*** | 0.75(0.57, 0.97)* |
| Richest | 5.11 | 1.27(1.23,1.30) | 1.27(1.23,1.30)*** | 0.94(0.66, 1.35) |
| Residence | | | | |
| Urban | 4.70 | Reference | Reference | |
| Rural | 3.68 | 1.00(0.99(1.021) | 1.02(0.99,1.04) | 1.42(1.10, 1.84)* |

**Key:** ANC = Antenatal care, NBR = negative binomial regression, ZINBR = Zero-inflated negative binomial regression # = number

* = significant at p-value less than 0.05

** = significant at p-value less than 0.01

*** = significant at p-value less than 0.001, AIRR = Adjusted incidence rate ratio AOR = adjusted odds ratio.

pregnancy because they have favorable knowledge about maternal and child health. One explanation would be that those women who have educated husbands were more likely to understand themselves due to support from their husbands which may increase discussion to get maternal health service utilization [53]. Therefore, to promote frequent antenatal care (ANC) visits, efforts should be made to improve maternal and husband's educational status.

More frequent ANC visits are directly associated with a higher number of household members. The frequencies of ANC among women were increased by 1% when the household size was increased. However, how the correlation exists between the number of household members and the number of ANC visits, a study conducted in Ethiopia supported this evidence [30]. The frequency of ANC visits among women decreased by 2.9% when the birth orders were increased. This is supported by the previous researches [36,54]. This may be due to increased confidence from prior birth experiences. When the number of under-five children increased, the frequency of ANC visits among women decreased by 2.7%. This is consistent with other study findings [55]. This could be due to mothers becoming overwhelmed with caring for their children at home, which may lead to reduced ANC utilization. The number of ANC visits directly and zero ANC visits are indirectly associated with higher levels of household wealth status as compared to women from the poorest household wealth index [11,17,56]. The possible reason may be due to the fact that the richest households will produce women who are more independent, educated, and confident to get maternal health services. In addition, a rich and educated community produces women who are more independent, educated, and confident to receive maternal health services. The number of ANC visits was higher among pregnant women who had wanted pregnancies whereas zero ANC visits was higher among pregnant women who had unwanted pregnancies and no more fertility desire. This finding was similar to previous study findings [52,57]. This may be explained because mothers whose pregnancies are desired have a better interest in satisfying themselves and getting maternal health services. It is obvious that if the pregnancy was wanted women's willingness to get health services would increase and there might be early detection of pregnancy, which in turn leads to early booking for ANC, as a result, the frequency of ANC visits would increase.

The odds of zero ANC visits are higher among women in rural residences as compared to urban residences. This finding was similar to previous study findings [18,58]. This may be

because rural or remote areas may have limited access to healthcare facilities, leading to no accessing antenatal care. Women living in these areas may face challenges in accessing services due to long distances or poor road infrastructure [59].

The strength of this study is the use of nationally representative data, which allows it to be generalizable to all Sub-Saharan African countries. This study relies on secondary data from the Demographic and Health Surveys (DHS), which may have limitations in terms of data accuracy and reliability due to the respondents' ability to accurately recall and report information. The study relies on self-reported data, which may be subject to recall bias or social desirability bias. Participants may over-report or under-report their ANC visits due to social expectations or memory limitations. The study design is cross-sectional, which limits the ability to establish causal relationships between the predictors and the number of ANC visits may also be a limitation of this study. We believe that the difference in health care professionals involved in conducting the ANC visit has an impact on the quality of ANC service and may have an impact on the number of ANC visits. In this study health care professionals involved in attending ANC visits during pregnancy like midwives, nurses, health officers, medical doctors, or obstetricians were not assessed due to the incompleteness of the data set may also be a limitation of the study. So we recommend that the researcher include this variable may help them understand of quality of ANC service and the number of ANC visits.

## Conclusion

The mean number of ANC visits among reproductive-age women in Sub-Saharan Africa is too lower than the new WHO recommendation of ANC visits for a positive pregnancy experience. This study also highlights that the proportion of at least eight ANC visits is low and there are still disparities in the mean number of ANC visits across different regions of SSA. The increasing maternal age, higher maternal and husband educational status, wanted pregnancy, the number of household members, the number of under-five children, and higher wealth index increase the number of ANC visits. Unwanted pregnancy, no more fertility desire, and rural residences were contributed for zero ANC visits in Sub-Saharan Africa. Therefore, efforts should be geared towards improving maternal and husband's educational status. We strongly recommend that the governments of SSA countries should empower women economically and educationally to achieve the goals of ANC as recommended by the WHO.

## Supporting information

**S1 File. The country included in the analysis with the recent round of DHS from 2017–2023.**
(DOCX)

## Acknowledgments

The authors are sincerely grateful to the Demographic Health Survey (DHS) program for providing us to use the DHS dataset through their archives https://dhsprogram.com/.

## Author Contributions

**Conceptualization:** Abel Endawkie.

**Data curation:** Abel Endawkie.

**Formal analysis:** Abel Endawkie.

**Investigation:** Abel Endawkie, Yawkal Tsega.

**Methodology:** Abel Endawkie, Natnael Kebede, Yawkal Tsega.

**Software:** Abel Endawkie.

**Supervision:** Abel Endawkie, Desale Bihonegn Asmamaw, Yawkal Tsega.

**Validation:** Abel Endawkie, Desale Bihonegn Asmamaw, Yawkal Tsega.

**Visualization:** Abel Endawkie, Yawkal Tsega.

**Writing – original draft:** Abel Endawkie.

**Writing – review & editing:** Abel Endawkie, Natnael Kebede, Desale Bihonegn Asmamaw, Yawkal Tsega.

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
