## [Decision Letter · Decision Letter 0]

21 Feb 2024

PONE-D-24-01064Predictors and number of Antenatal Care Visit among Reproductive Age Women in Sub-Saharan Africa Further Analysis of Recent DHS from 2017 -2023: Zero-Inflated Negative Binomial RegressionPLOS ONE

Dear Dr. Endawkie,

Thank you for submitting your manuscript to PLOS ONE. After careful consideration, we feel that it has merit but does not fully meet PLOS ONE’s publication criteria as it currently stands. Therefore, we invite you to submit a revised version of the manuscript that addresses the points raised during the review process.

We look forward to receiving your revised manuscript.

Kind regards,

Amos Buh, BSc., MPH, PhD

Academic Editor

PLOS ONE

Additional Editor Comments:

Please have a native English speaker review this manuscript for sentence and grammatical corrections. Also, carefully revise the manuscript following reviewers comments.

Reviewers' comments:

Reviewer's Responses to Questions

**Comments to the Author**

1. Is the manuscript technically sound, and do the data support the conclusions?

Reviewer #1: Partly

Reviewer #2: Yes

Reviewer #3: Yes

Reviewer #4: Yes

2. Has the statistical analysis been performed appropriately and rigorously? 

Reviewer #1: Yes

Reviewer #2: Yes

Reviewer #3: Yes

Reviewer #4: Yes

3. Have the authors made all data underlying the findings in their manuscript fully available?

Reviewer #1: Yes

Reviewer #2: Yes

Reviewer #3: No

Reviewer #4: Yes

4. Is the manuscript presented in an intelligible fashion and written in standard English?

Reviewer #1: No

Reviewer #2: Yes

Reviewer #3: No

Reviewer #4: Yes

5. Review Comments to the Author

Reviewer #1: Thanks for the opportunity to review this manuscript with the title “Predictors and number of Antenatal Care Visit among Reproductive Age Women in Sub-Saharan Africa Further Analysis of Recent DHS from 2017 -2023: Zero-Inflated Negative Binomial Regression” Where the aim was to determine mean number of ANC visit, proportion of at least eight ANC and its predictors among reproductive age women in Sub Saharan Africa using recent DHS data from 2017-2023 after the new WHO recommendation of ANC visit for a positive pregnancy experience.

General comments:

1. Mind a space before the references.

2. Try to use a clearer language so that sentences are not miss-understood. Clear out misspelling. For example “visits were more for women who have primary” It should be had instead of have. And higher frequency instead of “education had more frequency”. Read the whole text and correct the language.

Abstract

3. AIRR is written without an explanation

4. “were statistically significant with the numbers of ANC visits among reproductive age women in Sub-Saharan.” – write “associated with” instead of “were statistically significant with”

Introduction

5. This sentence is hard to understand “It is a sequence of clinical test and interventions that intention to make certain the well-being of each the mother and the fetus at some point of being pregnant (2, 3).” Please re-write.

Methods

6. Wo

7. In table 3 there are adjusted analyzes, however I can’t find in the text which variables they are adjusted for. Explain, and in addition explain your variables too .

8. How come you don’t need an ethical permit? It is written that the women were interviewed or do I misunderstand?

Results

9. The table 1 is hard to read. Std error doesn´t have to be included. Frequency and percentage can be one column as well as the upper and lower value of the confidence interval.

10. Try to condense the result, the most important numbers can be explained in the text, the rest is said by the table.

11. Row 227-230, comment the table, do not repeat it.

12. “Out of 188,880 women who were pregnant before the survey, 13579 (7.28%) had eight or more ANC visits between 2017 and 2023 in Sub-Saharan Africa (Table 2).” Do you mean ANC visits per pregnancy during the time of ANC during the whole time?

Discussion

13. “predictors of numbers of ANC visit” – How did you test which variables are predictors? Do you test the magnitude of the predictors?

14. Please re right: “One explanation would be that those women who have educated husbands will also likely to understand themselves and possibly this may due to support by their partner(husband)”

15. There are low number of ANC visits. Which are the consequences? Please tell us something about number of ANC and fetal outcome.

16. What could be done? What are your suggestions of action to increase number of ANC?

Reviewer #2: The identified spelling and grammatical errors have been highlighted appropriately in the relevant pages of the text for ease of reference and correction. Some sentences were unnecessarily long with repetitions and ambiguous. I have provided some corrections while leaving the ambiguous (unclear) ones for the Authors to resolve.

Reviewer #3: Good effort. However needs English language editing like in line 249. State limitations of study. List the subsaharan African nations involved in the study.. Authors may engage an English language editor.

Reviewer #4: Thank you for providing such an insightful research article. I appreciate the depth of your work. I would like to offer some constructive feedback and inquiries:

1- Have you analyzed the correlation between the number of antenatal care (ANC) visits and gestational age?

2- Could you elaborate on the antenatal care settings within each country included in the study? Additionally, it would be beneficial to elucidate any specific systems implemented to ensure patient attendance at these visits. Furthermore, providing clarity on the healthcare professionals typically involved in conducting these visits—whether it be midwives, nurses, or obstetricians—would enhance the understanding of the study's feasibility.

3- Have you measured the frequency of visits against the different medical conditions associated with pregnancy, and whether this pregnancy is low risk or high risk ?

4- Are there discernible reasons behind the less frequent ANC visits observed in your findings?.

5- Do you have any recommendations based on your study's findings?

6- Including a section on the limitations of your study would be instrumental in providing a comprehensive understanding of its scope and potential constraints.

6. PLOS authors have the option to publish the peer review history of their article (what does this mean?). If published, this will include your full peer review and any attached files.

Reviewer #1: No

Reviewer #2: **Yes: **Dr Francis E. Alu, Consultant ObGyn

Reviewer #3: **Yes: **Emmanuel Ajuluchukwu Ugwa

Reviewer #4: No

---

## [Author Response · Author response to Decision Letter 0]

24 Feb 2024

We feel all of you may understand our feeling and interest. we would like to say respond timely the required response

---

## [Decision Letter · Decision Letter 1]

10 Mar 2024

PONE-D-24-01064R1Predictors and Number of Antenatal Care Visit among Reproductive Age Women in Sub-Saharan Africa Further Analysis of Recent DHS from 2017 -2023: Zero-Inflated Negative Binomial RegressionPLOS ONE

Dear Dr. Endawkie,

Thank you for submitting your manuscript to PLOS ONE. After careful consideration, we feel that it has merit but does not fully meet PLOS ONE’s publication criteria as it currently stands. Therefore, we invite you to submit a revised version of the manuscript that addresses the points raised during the review process.

We look forward to receiving your revised manuscript.

Kind regards,

Amos Buh, BSc., MPH, PhD

Academic Editor

PLOS ONE

Additional Editor Comments:

Please respond to all reviewers' comments including those from the previous review which you ignored. Also, submit a rebuttal letter containing the responses to the reviewers' comments and highlight this on the main manuscript. I strongly suggest you have a native English speaker review this manuscript for grammatical errors.

Reviewers' comments:

Reviewer's Responses to Questions

**Comments to the Author**

1. If the authors have adequately addressed your comments raised in a previous round of review and you feel that this manuscript is now acceptable for publication, you may indicate that here to bypass the “Comments to the Author” section, enter your conflict of interest statement in the “Confidential to Editor” section, and submit your "Accept" recommendation.

Reviewer #1: All comments have been addressed

Reviewer #2: All comments have been addressed

Reviewer #3: (No Response)

Reviewer #4: (No Response)

2. Is the manuscript technically sound, and do the data support the conclusions?

Reviewer #1: Yes

Reviewer #2: Yes

Reviewer #3: Yes

Reviewer #4: (No Response)

3. Has the statistical analysis been performed appropriately and rigorously? 

Reviewer #1: Yes

Reviewer #2: Yes

Reviewer #3: Yes

Reviewer #4: Yes

4. Have the authors made all data underlying the findings in their manuscript fully available?

Reviewer #1: No

Reviewer #2: Yes

Reviewer #3: Yes

Reviewer #4: No

5. Is the manuscript presented in an intelligible fashion and written in standard English?

Reviewer #1: Yes

Reviewer #2: No

Reviewer #3: Yes

Reviewer #4: Yes

6. Review Comments to the Author

Reviewer #1: Thanks for your answers, corrections and changes. I'm satisfied with the answers and explanations.

Reviewer #2: Although most of the grammatical/spelling/typographical errors have been substantially addressed, some were either deliberately IGNORED or not NOTICED. Their correction would help make the article readable. For example, in the discussion section, line 249 reads: "The research was conducted a statistical analysis using ......" I corrected this to read "The research was conducted AS a statistical analysis using.... but this correction was ignored. Similarly, lines 288-291 starting with "To promote frequent antenatal care ........ " is ambiguous especially with the word "INTERN" and i suggested its revision(with a guide provided for this) but this was also IGNORED. Lines 325-331 was similarly AMBIGUOS and a suggestion made for its revision was IGNORED. It does not make intelligent reading as presently written and does not convey much. Line 128 under sampling method, the word "COMPLIED" was corrected to read "COMPILED" but this was IGNORED: In Table 1 under "WANTED (PREGNANCY TYPE) the word WNWANTED was corrected to UNWANTED but this was ignored! A suggestion for abbreviations to be written in full at first mention was also IGNORED. There are a few others which i chose not to point out here but which were highlighted in RED in my previous review and which are intended to make the English intelligible to read.

Reviewer #3: I am not aware that the authors have stated whether all 40 countries in sub saharan Africa wee involved inthe assessment. Authors should mention the countries within the subregion.

Reviewer #4: I couldn't find any evidence that the previous reviewer comments have been addressed. I suggest attaching a file containing the responses to the reviewers' comments and highlighting this on the main manuscript as well.

7. PLOS authors have the option to publish the peer review history of their article (what does this mean?). If published, this will include your full peer review and any attached files.

Reviewer #1: No

Reviewer #2: No

Reviewer #3: **Yes: **Emmanuel Ugwa PhD

Reviewer #4: No

---

## [Author Response · Author response to Decision Letter 1]

13 Mar 2024

Response for Reviewer #1: We sincerely appreciate your kindness and contribution to improving our manuscript. All the ideas you’ve raised are valuable for enhancing our work in the scientific context.

Response to Reviewer #2:

We sincerely apologize that the corrected version of our document did not reach you. Dear Reviewer, we deeply appreciate your insightful comments, which are invaluable for ensuring the scientific validity of our manuscript. We sincerely thank you for your patience and apologize for any inconvenience caused by the submission of our revised document. During the first revision submission process, where the revised version became intermixed with our initial document. Please rest assured that we made every effort to address your feedback and suggestions during the revision. Specifically, we incorporated your recommended changes to words and sentences, including the correction you pointed out in the sentence: For example “The research was conducted a statistical analysis using…” which we promptly adjusted to “The research was conducted as a statistical analysis using…” at the beginning of our discussion. Thank you once again for your valuable input.

Comment 1: For example, in the discussion section, line 249 reads: "The research was conducted a statistical analysis using ......" I corrected this to read "The research was conducted AS a statistical analysis using.... but this correction was ignored. 

Response 1: We deeply appreciate your insightful comments. In response to your suggestion, we deemed it important to make the sentence change. Accordingly, we revised and corrected it as it is. See line # 31

Comment2: Similarly, lines 288-291 starting with "To promote frequent antenatal care ........ " is ambiguous, especially with the word "INTERN" and i suggested its revision(with a guide provided for this) but this was also IGNORED. 

Response 2: We deeply appreciate your insightful comments. In response to your suggestion, we deemed it important to make the sentence change. Accordingly, we revised and corrected it as it is. See line # 325 

Comment 3: Lines 325-331 were similarly AMBIGUOS and a suggestion made for its revision was IGNORED. It does not make intelligent reading as presently written and does not convey much. 

Response 3: We deeply appreciate your insightful comments. In response to your suggestion, we deemed it important to make the sentence change. Accordingly, we revised and corrected it as it is. See line # 326-328

Comment4: Line 128 under sampling method, the word "COMPLIED" was corrected to read "COMPILED" but this was IGNORED: See fig1

Response 4: We deeply appreciate your insightful comments. In response to your suggestion, we deemed it important to make the sentence change. Accordingly, we revised and corrected as it is. As a reminder, we have made the necessary corrections. The figure has been submitted separately as “Fig1” in “TIF” format, and you can find it. See fig1

Comment5: In Table 1 under "WANTED (PREGNANCY TYPE) the word WNWANTED was corrected to UNWANTED but this was ignored! 

Response 5: We deeply appreciate your insightful comments. In response to your suggestion, we deemed it important to make the sentence change. Accordingly, we revised and corrected it as it is. You can find it in our revised version of the table in the manuscript. See table 1 and 2

Comment6: A suggestion for abbreviations to be written in full at first mention was also IGNORED. There are a few others which i chose not to point out here but which were highlighted in RED in my previous review and which are intended to make the English intelligible to read.

Response 6: We deeply appreciate your insightful comments. In response to your suggestion, we deemed it important to make the sentence change. Accordingly, we revised and corrected it as it is. You can find it in our revised version of the manuscript. 

Responses for Reviewr#3: We greatly appreciate your insightful comments. In response to your feedback, we initially attempted to include the specific country in our analysis during our first revision. In the second round revised version of our manuscript, we have included the countries within the sub-region (four regions of Africa) in the data source section of the method.

Response for Reviewer#4:We greatly appreciate your input. Honestly, we attached our response to all reviewers’ comments, which was saved as a file titled “Author Response to Reviewers’ Comments during the initial Revision. Now, we have included the author’s response to the reviewer’s comments in the revised manuscript below for your information.

Authors Response to the Reviewer Comment in the First Revision of Manuscript

Reviewer#1 Comments

Thanks for the opportunity to review this manuscript with the title “Predictors and number of Antenatal Care Visit among Reproductive Age Women in Sub-Saharan Africa Further Analysis of Recent DHS from 2017 -2023: Zero-Inflated Negative Binomial Regression” Where the aim was to determine mean number of ANC visit, proportion of at least eight ANC and its predictors among reproductive age women in Sub Saharan Africa using recent DHS data from 2017-2023 after the new WHO recommendation of ANC visit for a positive pregnancy experience.

Response: we are grateful to thank you for your Kindness and response. We have tried to correct your concern and suggestion thoroughly.

General comments:

Comment1: Mind a space before the references.

Response 1: We would like to thank you kindly for your deep comment. We had revised and corrected it.

 Comment2: Try to use a clearer language so that sentences are not miss-understood. Clear out misspelling. For example “visits were more for women who have primary” It should be had instead of have. And higher frequency instead of “education had more frequency”. Read the whole text and correct the language.

Response 2: We would like to thank you kindly for your deep comment. We have revised and corrected it. See line number 222

Response for general comment: We would like to thank you kindly for your deep comment. We have revised and corrected it.

Comment for Abstract

Comment3: AIRR is written without an explanation. 

Response 3: We would like to thank you kindly for your deep comment. We have revised and corrected it. See line number 31

Comment4: “were statistically significant with the numbers of ANC visits among reproductive age women in Sub-Saharan.” – write “associated with” instead of “were statistically significant with”. 

Response 4: We would like to thank you kindly for your deep comment. We have revised and corrected it. See line number 37

Comment for Introduction

Comment5: This sentence is hard to understand. “It is a sequence of clinical tests and interventions that intend to make certain the well-being of each mother and the fetus at some point of being pregnant (2, 3).” Please re-write. 

Response 5: We would like to thank you for your deep comment and we would like to say sorry for confusing you in writing. We have revised and corrected it. See line number 52-55

Comment for Methods

Comment6. Wo

Response 6: We really appreciate any concern you raise. However, we cannot understand the word (Wo)? We are open to receiving more elaboration of your concern if it is serious and do not hesitate to communicate.

Comment7. In table 3 there are adjusted analyses, however, I can’t find in the text which variables they are adjusted for. Explain, and in addition explain your variables too.

Response 7: We would like to thank you for your deep comment. We had written and included in the method of analysis. The adjusted analysis was conducted after Bivariable negative binomial regression analysis was conducted. Variables that had p-value<0.25 at Bivariable negative binomial regression analysis were selected for both multivariable standard and zero-inflated negative binomial regression. 

Comment8. How come you don’t need an ethical permit? It is written that the women were interviewed or do I misunderstand?

Response8: We would like to thank you for your deep comment. No ethical approval was needed because we had used the demographic and health survey which identifies all data before making it public, and the used DHS data sets are openly accessible at https://dhsprogram.com/. 

Comment: Results

Comment9. Table 1 is hard to read. Std error doesn´t have to be included. Frequency and percentage can be one column as well as the upper and lower value of the confidence interval.

Response9: We would like to thank you for your deep comment and we would like to apologize for confusing you in writing. We have revised and corrected it. See Table 1

Comment10. Try to condense the result, the most important numbers can be explained in the text, and the rest is said by the table.

Response10: We would like to thank you for your deep comment. We have revised and corrected it.

Comment11. Row 227-230, comment the table, do not repeat it.

Response11: We would like to thank you for your deep comment. We have revised and corrected it.

Comment12. “Out of 188,880 women who were pregnant before the survey, 13579 (7.28%) had eight or more ANC visits between 2017 and 2023 in Sub-Saharan Africa (Table 2).” Do you mean ANC visits per pregnancy during the time of ANC during the whole time?

Response 12: We would like to thank you for your deep comment and we would like to apologize for having confused you in writing. This is for interviewed reproductive-age women who were pregnant and gave birth five years before the survey (in general it is not per pregnancy but rather for the last pregnancy before the survey). 

Comment for Discussion

Comment13. “predictors of numbers of ANC visits” – How did you test which variables are predictors? Do you test the magnitude of the predictors?

Response 13: We would like to thank you for your deep comment. First for more clearer understanding of this study predictors were considered as factors/determinants for the numbers of ANC visits among reproductive-age women. The reason why we are using the term predictor is more appropriate for count model analysis than using associated factors determinants or covariates.

We had not conducted a predictive model analysis. We conducted a count model analysis to determine the mean number of ANC visits and factors for low number of ANC visits and higher number of ANC visits among reproductive women who were pregnant and gave birth before the survey. We measured the magnitude of each factor/predictor that contributed to ANC visit using count by incidence rate ratio. As you know incidence rate ratio (IRR) is a measure of association or effect measure which is used to measure the relationship between outcome and independent variable. 

 Comment514. Please re right: “One explanation would be that those women who have educated husbands will also likely to understand themselves and possibly this may be due to support by their partner (husband)”.

Response14: We would like to thank you for your deep comment and we would like to apologize for confusing you in writing. We have revised and corrected it.

Comment 15. There are low number of ANC visits. Which are the consequences? Please tell us something about the number of ANC and fetal outcomes.

Response 15: We would like to thank you for your deep comment. In this study, we determine the factors for low number of ANC visits and higher number of ANC visits among reproductive women who were pregnant and gave birth before the survey. We have not measured the consequence of the low number of ANC visits. However, the study we had conducted entitled ‘’Number of pregnancy loss and associated factors among reproductive age women who were ever pregnant before the survey in Sub-Saharan Africa using DHS data’’ showed that there is a high number of pregnancy loss among women who had the low number of ANC visit and poor quality of ANC. 

As you know in general, obviously a low number of ANC visits will result in poor fetal outcomes like abortion, stillbirth, low birth weight, and neonatal mortality. This is stated in the introduction part as the benefit, impact/ consequence of ANC on maternal and newborn health. 

Comment16. What could be done? What are your suggestions for action to increase the number of ANC?

Response 16: We would like to thank you for your deep comment. The possible recommendation to increase the number of ANC visits based on our findings:

1. Efforts should be geared towards improving maternal and husband’s educational status.

2. We strongly recommend that the governments of Sub-Saharan African countries should empower women economically and educationally to achieve the goals of ANC as recommended by the WHO.

Reviewer#2 Comment 

The identified spelling and grammatical errors have been highlighted appropriately in the relevant pages of the text for ease of reference and correction. Some sentences were unnecessarily long with repetitions and ambiguous. I have provided some corrections while leaving the ambiguous (unclear) ones for the Authors to resolve.

Response: We would like to thank you for your kind response and deep comment. We have revised and corrected it.

Reviewer#3 Comment

Good effort. However, I need English language editing like in line 249. State limitations of study. List the Sub-Saharan-African nations involved in the study. Authors may engage an English language editor.

Response: We would like to thank you for your kind response and deep comment. We have revised and corrected it. The study relies on secondary data from the Demographic and Health Surveys (DHS), which may have limitations in terms of data accuracy and reliability. The quality of the data depends on the collection methods and the respondents' ability to accurately recall and report information. The study relies on self-reported data, which may be subject to recall bias or social desirability bias. Participants may over-report or under-report their ANC visits due to social expectations or memory limitations.

The study design is cross-sectional, which limits the ability to establish causal relationships between the predictors and the number of ANC visits.

We had listed the countries included in this data analysis in the data source section of methods. See line number 102-103. 

Reviewer #4 comment: 

Thank you for providing such an insightful research article. I appreciate the depth of your work. I would like to offer some constructive feedback and inquiries.

Comment1: 1- Have you analyzed the correlation between the number of antenatal care (ANC) visits and gestational age?

Response 1: We would like to thank you for your kind response and deep comment. Since we had analyzed secondary data from publically available, the data set has no gestational age-related data because it was collected among reproductive-age women who were pregnant and gave birth five years before the survey. As a result, we did not analyze the correlation between the number of antenatal care (ANC) visits and gestational age. Our objective was to determine the mean number of ANC visits, the proportion of at least eight ANC visits, and its predictors among reproductive-age women in Sub-Saharan Africa using recent DHS data from 2017-2023 following the new WHO recommendation of ANC visits for a positive pregnancy experience.

Comment2- Could you elaborate on the antenatal care settings within each country included in the study? Additionally, it would be beneficial to elucidate any specific systems implemented to ensure patient attendance at these visits. Furthermore, providing clarity on the healthcare professionals typically involved in conducting these visits—whether it be midwives, nurses, or obstetricians—would enhance the understanding of the study's feasibility.

Response 2- We would like to thank you for your kind response and deep comment.

1. Yes, we had tried to elaborate in the result and discussion section As you see in Table 1 of the result there is the disparity of the average number of ANC visits in the regions of SSA. The mean number of ANC visits was lowest in Central Africa and highest in West Africa. These disparities could be due to inequalities in the accessibility of maternal health services, poor or absent transportation, inequality in the number of healthcare providers, and disparities

---

## [Decision Letter · Decision Letter 2]

14 Jun 2024

PONE-D-24-01064R2Predictors and Number of Antenatal Care Visits among Reproductive Age Women in Sub-Saharan Africa Further Analysis of Recent DHS from 2017 -2023: Zero-Inflated Negative Binomial RegressionPLOS ONE

Dear Dr. Endawkie,

Thank you for submitting your manuscript to PLOS ONE. After careful consideration, we feel that it has merit but does not fully meet PLOS ONE’s publication criteria as it currently stands. Therefore, we invite you to submit a revised version of the manuscript that addresses the points raised during the review process.

We look forward to receiving your revised manuscript.

Kind regards,

Amos Buh, BSc., MPH, PhD

Academic Editor

PLOS ONE

Additional Editor Comments:

During final checks of this article, concerns were raised that the statistical analyses may not have been evaluated in sufficient detail during peer review. To ensure that your work receives a thorough and rigorous review, the manuscript has now been assessed by one of our statistical advisors. Please accept our apologies that this review was not arranged at an earlier stage in the review process.

In addition to the reviewer comments, please note the following concerns:

1. The manuscript does not explicitly reference the DHS instrument(s) included in the analysis. Details of the specific countries and years should be included

2. The manuscript does not adequately cite and discuss in the Introduction any previous literature pooling DHS surveys to investigate predictors and number of antenatal care visits in sub-Saharan Africa. The manuscript should cite and discuss existing literature, and indicate how this study will build on the existing literature, in line with our 2nd publication criterion.

Reviewers' comments:

Reviewer's Responses to Questions

**Comments to the Author**

1. If the authors have adequately addressed your comments raised in a previous round of review and you feel that this manuscript is now acceptable for publication, you may indicate that here to bypass the “Comments to the Author” section, enter your conflict of interest statement in the “Confidential to Editor” section, and submit your "Accept" recommendation.

Reviewer #1: All comments have been addressed

Reviewer #2: All comments have been addressed

Reviewer #4: All comments have been addressed

Reviewer #5: (No Response)

2. Is the manuscript technically sound, and do the data support the conclusions?

Reviewer #1: Yes

Reviewer #2: Yes

Reviewer #4: Yes

Reviewer #5: Yes

3. Has the statistical analysis been performed appropriately and rigorously? 

Reviewer #1: Yes

Reviewer #2: Yes

Reviewer #4: Yes

Reviewer #5: No

4. Have the authors made all data underlying the findings in their manuscript fully available?

Reviewer #1: No

Reviewer #2: Yes

Reviewer #4: Yes

Reviewer #5: Yes

5. Is the manuscript presented in an intelligible fashion and written in standard English?

Reviewer #1: Yes

Reviewer #2: Yes

Reviewer #4: Yes

Reviewer #5: Yes

6. Review Comments to the Author

**Reviewer #1:** I have got resonably answers to my comments and these were adequate and enhanced the project futher.

**Reviewer #2: **The Authors have further addressed the issues i raised in my earlier review and effected the grammatical and typographical errors earlier highlighted. I recommend the acceptance of the manuscript for publication.

**Reviewer #4:** Thank you for your contribution to the current research field and your dedication towards addressing the previous reviewers' comments.

**Reviewer #5:** The authors studied variables associated with mean number of antenatal care among productive women Sub-Saharan Africa using data collected from 2017-2023 Demographic and Health Surveys (DHS).

I have some questions about the statistical analysis.

Did all pregnancies of the same women have separate entries in the database?

Due to the clustering of the data at several levels- pregnant women, household, sampling cluster, a multilevel mixed effects model should be used to account for the within clustering correlation.

More details are needed for the intra-class correlation coefficient calculation in lines 171-173. Please define sigma squared and pi squared. How were the values of sigma squared and pi squared obtained, estimates from a mixed model?

What variables were used as predictors for the zero component of the zero-inflated negative binomial model? Were any of the predictors significant?

Figure 1. What do the two numbers 191,344 and 188,880 stand for respectively?

Does “5 years before the survey” means “in the 5 years preceding the survey”?

7. PLOS authors have the option to publish the peer review history of their article (what does this mean?). If published, this will include your full peer review and any attached files.

Reviewer #1: No

Reviewer #2: No

Reviewer #4: No

Reviewer #5: No

---

## [Author Response · Author response to Decision Letter 2]

20 Jun 2024

Authors’ Responses to Editor and Reviewers Comments

Dear Academic Editor of Plose One 

 “Predictors and Number of Antenatal Care Visits among Reproductive Age Women in Sub-Saharan Africa Further Analysis of Recent DHS from 2017 -2023: Zero-Inflated Negative Binomial Regression”

Dear Plos One Editors and Reviewers;

We are thankful for your constructive comments. We have looked at the comments and have revised our paper accordingly. We hope our paper improved as a result of incorporating the reviewer’s and academic editor’s comments and suggestions. Here are the authors’ responses to the comments.

Please find for your kind consideration the following:

 A revised manuscript without track changes.

 A revised paper with tracked changes

 A rebuttal letter that responds to each point raised by the academic editor and reviewer. 

The point-by-point responses of authors are written by hoping these changes would meet with your favorable consideration, we are happy to hear if there are more comments and suggestions. Please do not hesitate to let us know if you have any questions.

Yours Sincerely 

Mr. Abel Endawkie Correspondence Author 

Department of Epidemiology and Biostatistics School of Public Health College of Medicine and Health Science Wollo University Dessie Ethiopia 

Tel. 251935459310 Email address abelendawkie@gmail.com

We have tried our best to improve it accordingly:

 Please revise the manuscript.

Point-by-point response of Authors for editor's and reviewers' comment 

Editor's comments to the Authors

Additional Editor Comments:

During final checks of this article, concerns were raised that the statistical analyses may not have been evaluated in sufficient detail during peer review. To ensure that your work receives a thorough and rigorous review, the manuscript has now been assessed by one of our statistical advisors. Please accept our apologies that this review was not arranged at an earlier stage in the review process.

Response: We sincerely appreciate your kindness, understanding, and prompt response. We kindly accept your apologies for any inconvenience caused by the delayed review process. Rest assured, our manuscript is now receiving the thorough evaluation it deserves. If you have any further questions or concerns, please don’t hesitate to reach out.

In addition to the reviewer comments, please note the following concerns:

Comment 1. The manuscript does not explicitly reference the DHS instrument(s) included in the analysis. Details of the specific countries and years should be included

Response 1: We are grateful to thank you for your kindness and response. We have tried to correct your concern and suggestion thoroughly. We have attached the details of the specific countries and years as supplementary files. 

Comment 2. The manuscript does not adequately cite and discuss in the Introduction any previous literature pooling DHS surveys to investigate predictors and number of antenatal care visits in sub-Saharan Africa. The manuscript should cite and discuss existing literature, and indicate how this study will build on the existing literature, in line with our 2nd publication criterion.

Response 2: We sincerely appreciate your kindness and prompt response. We have diligently addressed your concerns and suggestions

Authors Response to the Reviewer Comment 

Review Comments to the Author

Reviewer #1: I have got resonably answers to my comments and these were adequate and enhanced the project further.

Response: We sincerely appreciate your kindness and prompt response. Your valuable insights, comments, and suggestions for improving the scientific plausibility of our manuscript are highly appreciated. 

Reviewer #2: The Authors have further addressed the issues i raised in my earlier review and effected the grammatical and typographical errors earlier highlighted. I recommend the acceptance of the manuscript for publication.

Response: We sincerely appreciate your kindness and prompt response. Your valuable insights, comments, and suggestions for improving the scientific plausibility of our manuscript are highly appreciated.

Reviewer #4: Thank you for your contribution to the current research field and your dedication towards addressing the previous reviewers' comments.

Response: We sincerely appreciate your kindness and prompt response. Your valuable insights, comments, and suggestions for improving the scientific plausibility of our manuscript are highly appreciated.

Reviewer #5: The authors studied variables associated with mean number of antenatal care among productive women Sub-Saharan Africa using data collected from 2017-2023 Demographic and Health Surveys (DHS).

I have some questions about the statistical analysis.

Comment 1: Did all pregnancies of the same women have separate entries in the database?

Response 1: We sincerely appreciate your insightful comment and questions and we apologize for any confusion in our writing. Our study focuses on reproductive-age women who were pregnant and gave birth to determine the mean number of ANC vistas and its predictors during the last one/recent one pregnancy before the survey. Our study is not to each and all pregnancies before the survey but rather certainly we focused on the last one pregnancy before the survey.

Comment 2: Due to the clustering of the data at several levels- pregnant women, household, sampling cluster, a multilevel mixed effects model should be used to account for the within clustering correlation.

Response 2: We sincerely appreciate your insightful comment, and we apologize for any confusion in our writing. Before we conducted this analysis we had checked the cluster variation since the DHS data is hierarchical which assumes the pregnant women and their households are nested within the enumeration area (clusters). This suggests that pregnant women in households with similar characteristics can have different health outcomes when residing in different communities(clusters) with different characteristics. Therefore, to check whether there is distinct measures of variation and any potential clustering effect for conducting multilevel analysis: the intra-class correlation coefficient (ICC) was calculated, and ICC=3.2% which is less than 10% and even 5% which favors the individual level analysis using negative binomial regression analysis.

Comment 3: More details are needed for the intra-class correlation coefficient calculation in lines 171-173. Please define sigma squared and pi squared. How were the values of sigma squared and pi squared obtained, estimates from a mixed model?

Response 3: We sincerely appreciate your insightful comment, and we apologize for any confusion in our writing. Here is the explanation. The intra-class correlation coefficient (ICC) was calculated as ICC = δ2/(δ2+ π2/ 3) where δ2 indicates the estimated constant variance of clusters in the null model( obtained from variance components of the null model) and π2/ 3= 3.29 is the variance of the model with scale parameter.

Comment 4: What variables were used as predictors for the zero component of the zero-inflated negative binomial model? Were any of the predictors significant?

Response 4: We sincerely appreciate your deep, insightful, valuable comments and questions, and we apologize for any confusion in our writing. In the context of a zero-inflated negative binomial (ZINB) model, the selection of predictors for the zero component involves identifying factors that influence the probability of observing a zero count (excess zeros). The zero component of a ZINB model captures the excess zeros in the data. These zeros may arise due to a separate process (e.g., structural zeros) rather than the underlying count distribution.

As you know the selection of a predictor for the zero component of the ZINB model is based on considering potential predictors that might influence the excess zeros.

 In general, there are common predictors including Covariates like socio-demographic, individual-level, or environmental variables related to the study population. Contextual factors like region (place of residence), time, or other contextual information and Study-Specific Variables like Treatment group, and intervention. 

In our study, since it is secondary data collected using a cross-sectional study approach which has no follow-up time or specified intervention in the data.

 The selected predictors of zero component in our analysis were socio-demographic, individual-level factors, place of residence for zero component to account the excess zeros in the data.

Yes there are significant variable for zero ANC visit. From selected factors of zero component of zero-inflated negative binomial (ZINB) model, maternal educational and occupational status, husbands occupational status, unwanted pregnancy, no more fertility desire, media exposure, wealth index, and rural residences significantly associated with zero ANC visits either negatively or positively. 

Here is the command used for Zero-inflated negative binomial regression in Stata 17. “zinb m14(number of ANC) i.v013(maternal age ) i.v106(maternal educational status) i.respoworkings(maternal occupationalstatus) i.Haseduca(husband educational status) i.hasbanworkstatus_cat(husband educational status) i.wantedpreg(pregnancy type) i.fertilityprefernc_cat (fertility preference) i.mediaexposure(media exposure) birth order ( birth order) v136(household number) v137 (numbers of under-five child) i.v151(sex of household head) i.HHHage(age of household head) i.v190(wealth index) i.v025 (place of residence) [iweight =wt],inflate( i.v106(maternal educational status) i.respoworkings(maternal occupational status) i.Haseduca(husband educational status) i.hasbanworkstatus_cat(husband educational staus) i.wantedpreg(pregnancy type) i.fertilityprefernc_cat (fertility preference) i.mediaexposure(media exposure) birth order ( birth order) v136(household number) v137 (numbers of under-five child) i.v151(sex of household head) i.HHHage(age of household head) i.v190(wealth index) i.v025 (place of residence) irr iterate(300) nolog”

Comment 5: Figure 1. What do the two numbers 191,344 and 188,880 stand for respectively?

Response 5: We sincerely appreciate your insightful, and valuable questions. The 191,344 women mean the unweighted total sample size of screened (eligible) reproductive women obtained from the included dataset. The 188,880 indicates the weighted sample size of this study obtained by dividing those eligible reproductive women (V005) by 1,000,000 (V005/1000000) to account for unequal probability of sampling to reduce sample variability by the principle of the DHS program using sample weights before further analysis. 

Comment 6: Does “5 years before the survey” means “in the 5 years preceding the survey”?

Response 6: We are grateful to thank you for your kindness and response. We sincerely appreciate your deep, insightful, and valuable comment, and we apologize for any confusion in our writing. This study includes data collected in the DHS among reproductive-age women who were pregnant and gave birth in the last 5 years before the survey.

We feel all of you may understand our feelings and interests.

---

## [Decision Letter · Decision Letter 3]

2 Jul 2024

Predictors and Number of Antenatal Care Visits among Reproductive Age Women in Sub-Saharan Africa Further Analysis of Recent DHS from 2017 -2023: Zero-Inflated Negative Binomial Regression

PONE-D-24-01064R3

Dear Dr. Endawkie,

We’re pleased to inform you that your manuscript has been judged scientifically suitable for publication and will be formally accepted for publication once it meets all outstanding technical requirements.

Kind regards,

Amos Buh, BSc., MPH, PhD

Academic Editor

PLOS ONE

Additional Editor Comments (optional):

Reviewers' comments:

Reviewer's Responses to Questions

**Comments to the Author**

1. If the authors have adequately addressed your comments raised in a previous round of review and you feel that this manuscript is now acceptable for publication, you may indicate that here to bypass the “Comments to the Author” section, enter your conflict of interest statement in the “Confidential to Editor” section, and submit your "Accept" recommendation.

Reviewer #5: All comments have been addressed

2. Is the manuscript technically sound, and do the data support the conclusions?

Reviewer #5: Yes

3. Has the statistical analysis been performed appropriately and rigorously? 

Reviewer #5: Yes

4. Have the authors made all data underlying the findings in their manuscript fully available?

Reviewer #5: Yes

5. Is the manuscript presented in an intelligible fashion and written in standard English?

Reviewer #5: Yes

6. Review Comments to the Author

Reviewer #5: (No Response)

7. PLOS authors have the option to publish the peer review history of their article (what does this mean?). If published, this will include your full peer review and any attached files.

Reviewer #5: No

---

## [Editor Report · Acceptance letter]

18 Apr 2024

PONE-D-24-01064R2 

PLOS ONE

Dear Dr. Endawkie, 

I'm pleased to inform you that your manuscript has been deemed suitable for publication in PLOS ONE. Congratulations! Your manuscript is now being handed over to our production team.

Kind regards, 

on behalf of

Dr. Amos Buh 

Academic Editor

PLOS ONE